# Esophageal Cancer Stem-like Cells Resist Ferroptosis-Induced Cell Death by Active Hsp27-GPX4 Pathway

**DOI:** 10.3390/biom12010048

**Published:** 2021-12-29

**Authors:** Chen-Chi Liu, Hsin-Hsien Li, Jiun-Han Lin, Ming-Chen Chiang, Tien-Wei Hsu, Anna Fen-Yau Li, David Hung-Tsang Yen, Han-Shui Hsu, Shih-Chieh Hung

**Affiliations:** 1Division of Traumatology, Emergency Department, Taipei Veterans General Hospital, Taipei 112201, Taiwan; ccliu5@vghtpe.gov.tw (C.-C.L.); onetumorcell@hotmail.com (J.-H.L.); 650244@gmail.com (M.-C.C.); hjyen@vghtpe.gov.tw (D.H.-T.Y.); 2Faculty of Medicine, School of Medicine, National Yang Ming Chiao Tung University, Taipei 112304, Taiwan; fyli@vghtpe.gov.tw; 3Institute of Emergency and Critical Care Medicine, School of Medicine, National Yang Ming Chiao Tung University, Taipei 112304, Taiwan; hsinhsien@mail.cgu.edu.tw (H.-H.L.); tienvivi@hotmail.com (T.-W.H.); 4Department of Respiratory Therapy, College of Medicine, Chang Gung University, Taoyuan 33302, Taiwan; 5Division of Thoracic Surgery, Department of Surgery, Taipei Veterans General Hospital, Taipei 112201, Taiwan; 6Department of Pathology and Laboratory Medicine, Taipei Veterans General Hospital, Taipei 112201, Taiwan; 7Graduate Institute of New Drug Development, Biomedical Sciences, China Medical University, Taichung 404, Taiwan; 8Integrative Stem Cell Center, Department of Orthopedics, China Medical University Hospital, Taichung 404, Taiwan; 9Institute of Biomedical Sciences, Academia Sinica, Taipei 11529, Taiwan

**Keywords:** esophageal cancer, cancer stem cells, ferroptosis, Hsp27, GPX4

## Abstract

Cancer stem cells (CSCs), a subpopulation of cancer cells responsible for tumor initiation and treatment failure, are more susceptible to ferroptosis-inducing agents than bulk cancer cells. However, regulatory pathways controlling ferroptosis, which can selectively induce CSC death, are not fully understood. Here, we demonstrate that the CSCs of esophageal squamous carcinoma cells enriched by spheroid culture have increased intracellular iron levels and lipid peroxidation, thereby increasing exposure to several products of lipid peroxidation, such as MDA and 4-HNE. However, CSCs do not reduce cell viability until glutathione is depleted by erastin treatment. Mechanistic studies revealed that damage from elevated lipid peroxidation is avoided through the activation of Hsp27, which upregulates GPX4 and thereby rescues CSCs from ferroptosis-induced cell death. Our results also revealed a correlation between phospho-Hsp27 and GPX4 expression levels and poor prognosis in patients with esophageal cancer. Together, these data indicate that targeting Hsp27 or GPX4 to block this intrinsic protective mechanism against ferroptosis is a potential treatment strategy for eradicating CSC in esophageal squamous cell carcinoma.

## 1. Introduction

Esophageal cancer is among the most aggressive malignancies of the gastrointestinal tract and the sixth leading cause of cancer-related death, resulting in roughly 50,900 deaths worldwide in 2018 [1]. Despite the development of sophisticated multidisciplinary therapeutic strategies, the prognosis for esophageal squamous cell carcinoma (ESCC) is poor, with a 5-year overall survival rate of only 35% [1,2]. 

Recent research on anti-cancer therapies has been guided by the hypothesis that a small subpopulation of cancer cells possesses the properties of stem cells. Clinically, these cancer stem cells (CSCs) are responsible for resistance to conventional chemotherapy and radiotherapy, local recurrence, and metastasis. These CSC characteristics, including chemoresistance, tumor initiation, and epithelial–mesenchymal transition (EMT) have been demonstrated in esophageal squamous carcinoma cells enriched by spheroid culture [3].

Ferroptosis is morphologically, biochemically, and genetically distinct from other forms of cell death, including apoptosis, necrosis, and autophagy [4,5], which was first observed in Ras mutant tumor cells treated with oncogenic Ras-selective lethal small molecules (referred to as ferroptosis inducers), such as erastin and RSL3 [6]. The former inactivates glutathione peroxidase (GPX) through glutathione (GSH) depletion, while the latter binds to and inactivates GPX4 [7]. GPX4 is the only enzyme capable of reducing lipid hydroperoxides within biological membranes. This enzyme catalyzes GSH-dependent reduction and vigorously inhibits ferroptosis. Recent research has also demonstrated that cancer cells in a highly mesenchymal state are highly sensitive to the inhibition of GPX4 [8,9], which means that CSCs might be more susceptible to ferroptosis-inducing agents than bulk cancer cells. However, the regulatory pathways controlling ferroptosis, which can selectively induce CSC death, are not fully understood. 

Ferroptosis is referred to as an iron-dependent process since the accumulation of lipid ROS and cell death induced by erastin can be suppressed by iron chelator deferoxamine (DFO). Erastin-induced ferroptosis is positively correlated with the availability of intracellular iron [10]. Nonetheless, the specific role of iron in ferroptosis remains unclear. Recent evidence suggests that iron contributes to tumor initiation and progression on the basis of the fact that the metabolic demand for iron is stronger in cancer cells than in normal cells [11,12]. A number of studies have revealed elevated iron levels in CSCs in lung cancer and cholangiocarcinomas [13,14]. Note, however, that excess iron levels can lead to ROS formation via the Fenton reaction followed by cell death via ferroptosis, even in cancer cells or CSCs. The mechanism by which CSCs protect themselves against ROS and ferroptosis in an iron-rich environment has yet to be elucidated. 

Hsp27 is a 27-kDa protein produced by cells exposed to stressful conditions, such as heat shock. In cancer research, Hsp27 expression has been observed in various types of cancers and has been linked to chemotherapy resistance. In a previous study, we demonstrated that the expression of phospho-Hsp27 by tumors is predictive of a poor prognosis in cases of esophageal cancer [3]. In the current study, we investigated the reprogramming of iron metabolism in esophageal CSCs and the mechanism underlying this process. Our investigation was predicated on the hypothesis that esophageal CSCs differ from bulk cancer cells in their demand for iron and that Hsp27 plays a role in protecting CSCs from ferroptosis. Our results revealed elevated iron levels and lipid peroxidation in esophageal CSCs. We also determined that Hsp27 upregulates xCT-GPX4 through the inhibition of p53 to protect CSCs from ferroptosis. Finally, we determined that Hsp27 and GPX4 are valuable prognostic indicators in cases of esophageal cancer. 

## 2. Materials and Methods

### 2.1. Cell Line Culture and Reagents

Human esophageal squamous carcinoma cell lines CE81T [15] and TE1 [16], which have been established in the countries with a high incidence of esophageal cancer, were used in the study. Cells were grown in Dulbecco modified Eagle medium (DMEM) (Corning, Manassas, VA, USA) containing 10 units/mL penicillin, 10 μg/mL streptomycin, 2 mM glutamine, and 10% fetal bovine serum (Corning, Manassas, VA, USA) in a 37°C humidified atmosphere with 5% CO2. For enrichment of CSCs in spheroid culture, esophageal cancer cells were suspended in a tumorsphere medium consisting of DMEM/F12 (Corning, Manassas, VA, USA), N2 supplement (Gibco Life Technologies, Paisley, UK), human recombinant epidermal growth factor (EGF) (20 ng/mL, PeproTech, Rocky Hill, NJ, USA), and basic fibroblastic growth factor (bFGF) (10 ng/mL, PeproTech, Rocky Hill, NJ, USA) in the ultra-low dish (Corning, Manassas, VA, USA). Spheres were defined as cell colonies >50 µm in diameter and >50% in an area showing a 3-dimensional structure and blurred cell margins. Sphere formation was observed beginning on day 1 of culture. Cells were centrifugated at a low speed (1000 rpm) and washed with PBS twice to harvest spheroid cells, and protein lysates were collected on day 7 of culture for all experiments. The treatment reagents included erastin (20 µM, Selleckchem, TX, USA), ferrostatin-1 (1 µM, Selleckchem, TX, USA), liproxstatin-1 (1 µM, Selleckchem, TX, USA), ZVAD-FMK (50 µM, Sigma, St. Louis, MO, USA), necrostatin-1 (50 µM, Sigma, St. Louis, MO, USA) for 24 h.

### 2.2. Quantitative Real-Time PCR

An illustraTM RNAspin Mini (GE Healthcare, Little Chalfont, UK) was used to extract total RNA, and RNA was reverse-transcribed using SuperScript III RT (Invitrogen Life Technologies, Carlsbad, CA, USA) according to the manufacture’s specifications. For the real-time PCR, we used SensiFAST SYBR Hi-ROX Mix (Lot no: SFSH-717111A, Bioline, London, UK) and a StepOne Plus Real-Time PCR system (Applied Biosystems, Darmstadt, Germany) with specific primers. The results were calculated using the ∆∆CT equation and were expressed as multiples of change relative to a control sample. The primers used are as follows: GAPDH (forward: 5′-CAACTACATGGTTTACATGTTC-3′, reverse: 5′-GCCAGTGGACTCCACGAC-3′); Nanog (Forward: 5′-GTCCCGGTCAAGAAACAGAA-3′, reverse: 5′-TGCGTCACACCATTGCTATT-3′); Sox2 (forward: 5′-ATGGGTTCGGTGGTCAAGT-3′, reverse: 5′-ATGTGTGAGAGGGGCAGTGT-3′); Oct4 (forward: 5′-ATTCAGCCAAACGACCATCT-3′, reverse: 5′-ACACTCGGACCACATCCTTC-3′).

### 2.3. Iron Assay

The Iron Assay Kit (Colorimetric) (ab83366, Abcam, Cambridge, MA, USA) provides a simple, convenient means of measuring Ferrous and/or Ferric ions in samples. The ferric carrier protein dissociates ferric into solution in the presence of acid buffer. After reduction to the ferrous form (Fe^2+^), iron reacts with Ferene S (an iron chromogen) to produce a stable colored complex and give absorbance at 593 nm. A specific chelate chemical is included in the buffer to block copper ion (Cu^2+^) interference.

### 2.4. Determination of the Labile Iron Pool (LIP) 

LIP was determined according to a previous study [17]. In brief, cells and spheres were washed twice with PBS and centrifuged for 10 min at 1000 rpm. After the removal of the supernatant, spheres were treated with 500 μL of Accutase (Corning, Manassas, VA, USA), incubated for 5 min at 37 °C, and the wash process was repeated. Then, Calcein-AM 0.25 µM was added, and the solution was incubated for 15 min at 37 °C, washed with PBS, and centrifuged for 5 min at 1000 rpm. Afterward, the solution was treated Trypan blue (1 μL) and 1 mL of Calcein-AM buffer for 3–5 min and centrifuged for 5 min at 1000 rpm. After removing the supernatant, Desferrioxamine mesylate salt (DFO, D9533, Sigma-Aldrich, St. Louis, MO, USA) 100 uM was added, and the solution was covered to block light and incubated for 30 min at 37 °C. Cells were analyzed by a flow cytometer (FACS-Calibur^®^, Becton-Dickinson, Immunofluorometry Systems, Mountain View, CA, USA).

### 2.5. Western Blot Analysis

Cell extracts were incubated in a commercial lysis buffer (Pierce, Rockford, IL, USA) plus protease inhibitor cocktail (Pierce, Rockford, IL, USA), and the protein concentration was determined using the bicinchoninic acid (BCA) assay (Pierce, Rockford, IL, USA). Protein samples were separated on SDS–PAGE and transferred onto PVDF membranes, followed by blocking with 5% milk in TBST (20 mM Tris–HCl [pH 7.2], 137 mM NaCl, 1% Tween 20). Membranes were then probed with the indicated primary antibodies, followed by corresponding secondary antibodies, and detected using a chemiluminescence assay (PerkinElmer Life and Analytical Sciences, Boston, MA, USA). Membranes were exposed to X-ray film to visualize the bands (Amersham Pharmacia Biotech, Piscataway, NJ, USA). 

The primary antibodies against TfR1 (1:1000; cat. no. tcea17307), FTL (1:2000; cat. no. tcea16584), FTH (1:2000; cat. no. tcua2654), and FPN (1:1000; cat. no. tcua8598) were purchased from Taiclone; IRP1 (1:100; cat. no. sc-166022) and IRP2 (1:100; cat. no. sc-33682) were purchased from Santa Cruz. HSP27 (1:3000; cat. no. ARG65757) and Phospho-HSP27 (Ser78) (1:1000; cat. no. 2405S) were purchased from Cell Signaling. P53 (1:2000; cat. no. ab131442), xCT (1:2000; cat. no. ab111822), GPX4 (1:10,000; cat. no. ab40993), and Ferroportin (1:1000; cat. no. ab78066) were purchased from Abcam; Beta Actin (1:10,000; cat. no. 20536-1-AP) was purchased from Proteintech (Peprotech, Rocky Hill, NJ, USA).

### 2.6. Malondialdehyde (MDA) Assay 

The relative MDA concentration in cell lysates was assessed using a Lipid Peroxidation Assay Kit (ab118970, Abcam, Cambridge, MA, USA) according to the manufacturer’s instructions. Briefly, the MDA in the sample was reacted with thiobarbituric acid (TBA) to generate an MDA–TBA adduct. The MDA–TBA adduct was quantified colorimetrically (OD532 nm).

### 2.7. 4-Hydroxynonenal (4-HNE) Assay

The 4-HNE levels were measured using a 4-HNE ELISA kit (Elabscience Biotechnology, Bethesda, MD, USA), according to the manufacturer’s instructions. In brief, the cell pellets were washed and centrifuged for 10 min at 1500× *g* at 2–8 °C. After removal of the cell fragments, the supernatant was collected to carry out the assay. We added 50 μL standard or sample with 50 μL Biotinylated Detection antibody working solution to each well and incubated the, for 45 min at 37 °C. Then, we added 350 μL of wash buffer, soaked for 1 min, and repeated 3 times. Subsequently, 100 μL of HRP Conjugate working solution was added to each well, and the plate was incubated for 30 min at 37 °C. The process was repeated 5 times. After that, we added 90 μL of Substrate Reagent, covered the plate to block light, and incubated it for 15 mins at 37 °C. Finally, we added 50 μL of Stop Solution and read the plate at 450 nm using a microplate reader (Spectramax iD5 multi-mode microplate reader, Molecular Devices, Sunnyvale, CA, USA).

### 2.8. Glutathione Measurement

We used the Glutathione (GSSG/GSH) Detection Kit (Enzo Life Sciences, Farmingdale, NY, USA) and followed the product instructions to determine GSH levels. Briefly, the GSH assay buffer and supernatant sample were mixed, and the absorbance in the wells was recorded at 405 nm or 414 nm using a plate reader at 1 min intervals over a 10 min period. Reduced GSH = Total glutathione-Oxidized GSSG.

### 2.9. C11-BODIPY 581/591 Staining 

C11-BODIPY (581/591) was used to detect ROS in cells and membranes. The oxidation of the polyunsaturated butadienyl portion of the dye resulted in a shift of the fluorescence emission peak from 590 nm (red) to 510 nm (green). For flow cytometry, cells were stained with 5 μM C11-BODIPY (581/591) for 30 min at 37 °C and analyzed using a flow cytometer.

### 2.10. Cell Viability Assay (CCK-8 Assay)

The cell viability was evaluated using the Cell Counting Kit-8 (CCK-8) (CK04-13, Dojindo Laboratories, Rockville, MD, USA) according to the manufacturer’s instructions. We initially seeded 10^6^ cells in the spheroid medium in an ultra-low dish to form spheres. Spheres and cells were harvested, centrifuged, and counted, and 2 × 10^5^ cells were resuspended in 1 mL of serum-free DMEM medium. Then, we mixed 50 μL of the cell suspension fluid and 100 μL of serum-free DMEM medium in a 96-well plate. Bulk cancer cells were also seeded on 96-well plates at a density of 10^4^ cells per well. After drug treatment, 10 μL of CCK-8 solution was added to each well at 37 °C for 3 h, and then the plate was read at 450 nm using a microplate reader.

### 2.11. Tumor Xenograft Mouse Model and Treatment

The study protocols involving mice were approved by the Institutional Animal Committee of Taipei Veterans General Hospital (IACUC no.: 2019-145). The male NOD/SCID mice purchased from BioLASCO Experimental Animal Center (Taiwan Co., Ltd., BioLASCO, Taipei, Taiwan) were maintained in specific pathogen-free conditions. The mice were used for experiments at 6–8 weeks of age. For tumor volume measurement after the erastin experiment (Figure 3G,H), mice underwent subcutaneous injection of 5 × 10^5^ CE81T cells per injection site to form a tumor. Because the tumors formed by TE1 cells were too small, we used CE81T cells to perform this experiment. About 13 days after injection, the tumor became palpable, with a size approaching 50 mm^3^. After tumor formation, mice were intraperitoneally injected with erastin (30 mg/kg, cat. no. S7242, Selleckchem, TX, USA) every other day for 2 weeks. Tumor size (length and width) was measured thrice a week. Tumor volumes were calculated as follows: tumor volume = (length × width^2^)/2. Mice were sacrificed at the end of the experiment, and the tumors were harvested for photographs and weight measurement.

### 2.12. Lentiviral Vector Production and Cell Infection

The shRNA expression plasmids for Hsp27 (TRCN0000008753, TRCN0000011466) were provided by the RNAi core facility, Academia Sinica. Subconfluent cells were infected with lentivirus in the presence of 8 μg/mL polybrene (Sigma-Aldrich, St. Louis, MO, USA). At 24 h post-infection, media were removed and replaced with fresh growth media containing puromycin (4 μg/mL) to select for infected cells after 48 h post-infection. 

### 2.13. Immunohistochemistry

The use of human tissues or tumor specimen samples was approved by the IRB of Taipei Veterans General Hospital (IRB no.: 2019-10-010BC). Patients who underwent surgical resection for esophageal cancer were enrolled in this study. None of these patients received neoadjuvant chemotherapy or radiotherapy. The clinical data were collected by chart review. Paraffin blocks of tumor specimens were cut into 4 μm slices, processed using standard deparaffinization and rehydration techniques, and then subjected to hematoxylin (3801522, Leica Surgipath^®^, Leica Biosystems, Wetzlar, Germany) and eosin (3801602, Leica Surgipath^®^, Leica Biosystems, Wetzlar, Germany) staining and IHC analysis. Antibodies against phospho-Hsp27 (1:50; cat. no. 2405S, Cell Signaling Technology, Danvers, MA, USA) and GPX4 (1:250; cat. no. ab125066, Cell Signaling Technology, Danvers, MA, USA) were used as the primary antibodies to detect the protein expression. The cell nuclei were counterstained with hematoxylin. Images were acquired with the Olympus BX43 microscope to assess the proportion of positively stained cells. The score of positive tumor cell frequency and reaction intensity were used together to determine the result of the IHC analysis. The scores for positive tumor cell frequencies were: 0, <25%; 1, 25–50%; 2, 50–75%; 3, >75%. The scores for reaction intensity were: 0, no reaction; 1, weak; 2, moderate; 3, strong. The total score equaled the score of the positive tumor cell frequency added to the score of the reaction intensity. If the total score was more than two, we determined the expression of IHC was positive. On the contrary, if the total score was one or zero, the expression of IHC was negative.

### 2.14. Statistical Analysis 

Values are shown as the means ± standard deviation of the mean of measurements of at least three independently performed experiments to avoid possible variation of cell cultures. Quantitative analysis of all Western blots was carried out using Image-J software. Student’s *t*-test or two-way ANOVA was employed. For the clinical study, the correlations between immunohistochemical results and clinicopathological variables were analyzed by Pearson’s Chi-square test. Survival curves were calculated using the Kaplan–Meier method, and comparisons were performed using the log-rank test. *p* < 0.05 was considered statistically significant.

## 3. Results

### 3.1. Elevated Iron Content and Intracellular Iron in Esophageal CSCs 

To investigate whether the iron content in esophageal CSCs was different from the bulk cancer cells, we enriched CSCs from two human ESCC cell lines, TE1 and CE81T, via spheroid culture for 7 days, and then compared their iron content and the ratio of ferrous (Fe^2+^) and ferric (Fe^3+^) ions with the cells in adherent culture (control). The capacity of spheroid culture to enrich ESCC cell lines was confirmed by an increase in the expression of pluripotent genes, including Oct4, Nanog, and Sox2 (Appendix A). Esophageal CSCs increased in iron content, particularly in terms of ferrous (Fe^2+^) iron, as assayed with an iron assay kit (Figure 1A). The labile iron pool (LIP), indicating the most intracellular free ferrous iron, was also increased in esophageal CSCs (Figure 1B). Furthermore, we examined iron regulatory proteins 1 and 2 (IRP1 and IRP2), the mammalian proteins that control cellular iron balance and are regulated by the cytosolic iron level. Western blot analysis showed that, compared with bulk cancer cells, the protein level of IRP2 but not IRP1 was greatly reduced in esophageal CSCs (Figure1C), reflecting the iron-replete status of CSCs, which may make IRP2 protein unstable and inactivate IRP1 [18]. A decreased IRP2 level or inactivated IRP1 would subsequently induce translational activation of ferritin [18] Consistently, Western blot analysis showed that esophageal CSCs had higher levels of ferritin heavy chain (FTH) and ferritin light chain (FTL) compared with bulk cancer cells (Figure 1C). Interestingly, the expression of ferroportin (FPN) was reduced despite high intracellular iron status in esophageal CSCs. This finding is similar to that in the intestinal epithelial cells of animals fed with high iron-diets [19,20]. This is attributed to the fact that FPN transcripts in intestinal epithelial cells are not sensitive to iron levels because of the lack of 5′-IRE [21], although this has not been studied in esophageal CSCs. These results indicate a unique feature of iron metabolism in esophageal CSCs in which reduced iron export leads to increased intracellular iron content, which leads to IRP2 degradation and IRP1 inactivation and subsequent FTH and FTL upregulation.

### 3.2. Elevated Lipid Peroxidation in Esophageal CSCs 

Elevated levels of intracellular iron should lead to ROS formation via the Fenton reaction with a corresponding increase in lipid peroxidation, resulting in ferroptosis [22]. Thus, we examined the end products of lipid peroxidation, including malondialdehyde (MDA) and 4-hydroxynonenal (4-HNE). We also used C-11 BODIPY staining for the detection of ROS in the cell membrane. Intriguingly, we observed higher MDA and 4-HNE levels in esophageal CSCs than in bulk cancer cells (Figure 2A,B). We also observed higher C-11 BODIPY levels in esophageal CSCs (Figure 2C). Nonetheless, the CSCs appeared resistant to lipid peroxidation and actually presented the characteristics of tumor initiation and chemoresistance [3]. These data suggest that CSCs possess a protective mechanism against damage caused by elevated iron content and lipid peroxidation.

### 3.3. Upregulation of GPX4 and xCT to Protect Esophageal CSCs from Ferroptosis

Ferroptosis is triggered mainly by a reduction in the detoxification of lipid peroxides by GPX4. Thus, the elevated lipid peroxidation levels observed in esophageal CSCs prompted an investigation of the enzymatic activity by GPX4. Compared with bulk cancer cells, GPX4 was upregulated in esophageal CSCs (Figure 3A). The ratio of GSH to GSSH indicated the enzymatic activity of GPX4 was also increased in esophageal CSCs (Figure 3B). System Xc- (xCT), the main cellular importer of cystine for GSH synthesis [23], was also upregulated in esophageal CSCs (Figure 3A). Erastin has a strong inhibitory effect on xCT [6] and can inhibit the activity of certain GSH-related enzymes, such as GPX4, thereby causing more deadly oxidative damage to cells [7]. Erastin treatment reduced the cell viability of CSCs and bulk cancer cells in a dose-dependent manner, and at higher concentrations, such as 10 and 20 µM, erastin has a more pronounced effect on CSCs than bulk cancer cells (Appendix A). Furthermore, erastin-induced cell death was alleviated by ferroptosis inhibitors, ferrostatin-1 [6], and liproxstatin-1 [24], but it was unaffected by the inhibitors of apoptosis (ZVAD-FMK) or necrosis (necrostatin-1) [6] (Appendix A), suggesting that erastin-induced cell death in CSCs was mediated by ferroptosis alone. Erastin also increased MDA and 4-HNE in esophageal CSCs compared with bulk cancer cells (Figure 3D,E). Furthermore, C-11 BODIPY staining revealed that erastin increased the level of membranous ROS in esophageal CSCs (Figure 3F). These in vitro data indicate that the upregulation of xCT and GPX4 in esophageal CSCs is necessary to resist elevated lipid peroxidation levels.

To investigate the effect of xCT inhibition in CSCs in vivo, we used a xenograft tumor model with subcutaneous injection of CE81T cells that were cultivated in spheroid culture or adherent culture. After the subcutaneous tumor was palpable (about 13 days after cells injection), erastin (40 mg/kg) was intraperitoneally injected twice every other day for 2 weeks [25]. Compared with tumors grown from adherent cultured cells, erastin significantly reduced the volume and weight of tumors formed by CE81T cells in spheroid culture (Figure 3G,H). These results indicate that targeting xCT and GPX4 might help eradicate CSCs.

### 3.4. Hsp27 Upregulated xCT/GPX4 by Inhibiting p53 to Reduce Lipid Peroxidation

Our previous study [3] demonstrated that ESCC cell lines in spheroid culture exhibit the properties of CSCs, and Hsp27 was shown to be a crucial element in maintaining the esophageal CSC phenotype. Since the inhibition of xCT/GPX4 induces more ferroptosis in esophageal CSCs, we investigated the relationship between Hsp27 and several known key regulators of ferroptosis, including p53. Western blot analysis revealed that the phosphorylation of Hsp27 and xCT-GPX4 were upregulated and p53 was downregulated in esophageal CSCs (Figure 4A, left panel). To investigate the role of Hsp27 in protecting against damage induced by an endogenous increase in iron content and corresponding lipid peroxidation, we performed knockdown of Hsp27 in ESCC cell lines in spheroid culture and examined the resulting changes. Hsp27 knockdown was shown to induce an upregulation of p53 and a downregulation of xCT-GPX4 (Figure 4A, right panel). These findings suggest that Hsp27 serves upstream in the p53-xCT-GPX4 signaling pathway in esophageal CSCs. In addition, the enzymatic activity of GPX4 was reduced in cells with Hsp27 knockdown in spheroid culture (Figure 4B). MDA and 4-HNE were also increased after Hsp27 knockdown (Figure 4C,D). Together, these findings suggest that Hsp27 is required to diminish lipid peroxidation in esophageal CSCs.

### 3.5. Expression of Phospho-Hsp27/GPX4 Is Predictive of Poor Prognosis in Cases of Esophageal Cancer

To investigate the clinical significance of Hsp27and GPX4, we utilized immunohistochemical (IHC) staining to examine the expression of Hsp27 and GPX4 in the tumorous specimens of patients who underwent surgical resection for esophageal cancer in our hospital. The clinical demographics of these patients are listed in Appendix A. Kaplan–Meier analysis revealed that patients who displayed positive staining of either phospho-Hsp27 or GPX4 (Figure 5A) in their tumors exhibited poorer survival compared with patients who were negative for these markers (Figure 5B). The survival difference in the double-positive versus double-negative expression of phospho-Hsp27 and GPX4 was more obvious than that in single-positive versus single-negative groups (Figure 5B). These findings suggest that phospho-Hsp27 and GPX4 may be valuable markers to predict the prognosis of patients with esophageal cancer.

## 4. Discussion

Esophageal cancer is characterized by a high incidence of disease recurrence and an extremely poor prognosis, despite surgical resection and adjuvant therapies. This aggressive clinical course can be explained by the theory of CSC. In the current study, our use of a serum-free culture to enrich esophageal squamous carcinoma cells revealed elevated iron levels (particularly intracellular free iron) and corresponding lipid peroxidation in esophageal CSCs. Under these conditions, CSCs should die from ferroptosis; however, Hsp27 activation protects CSCs from ferroptosis by upregulating xCT and GPX4. These findings elucidate the intrinsic self-protecting mechanism of CSCs in preventing ferroptosis, and more importantly, these results are clinically applicable. We demonstrated that the overexpression of phospho-Hsp27and GPX4 in tumor specimens was associated with a poor prognosis in patients with esophageal cancer. Patients who presented an overexpression of both phospho-Hsp27 and GPX4 faced an even worse prognosis. This was the first study to demonstrate the use of GPX4 as a prognostic indicator in esophageal cancer. 

Iron is an essential nutrient for a variety of critical cellular functions, such as oxygen transport, cell respiration, and DNA synthesis. Iron deficiency can lead to arrested cell growth or death; however, iron overload can also be toxic, resulting in an increase in ROS levels. Cancer cells require more iron than do normal cells because of their rapid proliferation and growth. Notably, upregulation of iron storage as well as downregulation of iron export has been associated with several types of cancer [11]. FPN1 expression is diminished in breast, prostate, and hepatocellular cancer cells compared with normal cells, and low FPN1 expression levels in breast cancer have been linked to a better prognosis [26]. Theoretically, CSCs might have different iron demands and metabolism because of their slower proliferation rate and higher metabolic rate compared with bulk cancer cells. Raggi et al. reported decreased FPN expression as well as increased FTH expression in the CSCs of cholangiocarcinoma [14]. ROS production is increased in CSCs than in ordinary cholangiocarcinoma cells. In a genetic model of ovarian tumor-initiating cells (TIC), Basuli et al. reported variations in iron metabolism with lower FPN expression levels as well as higher LIP [27]. These findings present the same trend observed in the esophageal CSCs in our current study. We not only addressed the change in the iron metabolism profile; we also investigated iron content, intracellular free iron, and the redox status in CSCs. Overall, we demonstrated an elevated iron content and lipid peroxidation in CSCs. The decreased export of iron might lead to increased iron content and high metabolic available iron in LIP, which results in an increase in iron storage in esophageal CSCs. The ferric (Fe^3+^) iron stored in ferritin is stable and nontoxic; however, elevated ferrous (Fe^2+^) iron levels in LIP tend to produce hydroxyl radicals (·OH) via the Fenton reaction with a corresponding increase in the production of polyunsaturated fatty acid-containing phospholipid hydroperoxides, which could induce ferroptosis. 

CSCs have an intrinsic mechanism to inhibit lipid peroxidation and thereby avoid ferroptosis under elevated iron content. GPX4 is the only enzyme capable of reducing lipid hydroperoxides within the cell membrane [28]. Viswanathan et al. demonstrated that cancer cells existing in a high-mesenchymal state are highly sensitive to GPX4 inhibition [8,9]. Peng et al. also described the role of GPX4 in stemness phenotype maintenance and oxidative homeostasis in pancreatic CSCs [29]. Few reports have discussed the role of GPX4 in CSCs; however, it should be possible to predict the upregulation of GPX4 in response to increased redox status. Our results revealed the upregulation of GPX4 in esophageal CSCs and clarified the upstream changes in GPX4, including xCT, p53, and Hsp27. We further demonstrated that the expression of GPX4 in tumorous specimens from esophageal cancer patients is correlated with a poor prognosis (Figure 5A,B). Our findings also support recent studies that identified ferroptosis-related gene signatures, which can be used to predict the prognosis and immunological responses of esophageal squamous cell carcinoma and other cancers [30,31]. 

## 5. Conclusions

In conclusion, esophageal CSCs have increased iron content and lipid peroxidation. Hsp27 is activated to upregulate GPX4 to protect esophageal CSCSs from ferroptosis. More importantly, we determined that the expression of phospho-Hsp27 and GPX4 in tumorous specimens is predictive of poor prognosis in patients with esophageal cancer. Targeting Hsp27 or GPX4 could be a novel strategy for eradicating CSCs.

## Figures and Tables

**Figure 1 biomolecules-12-00048-f001:**
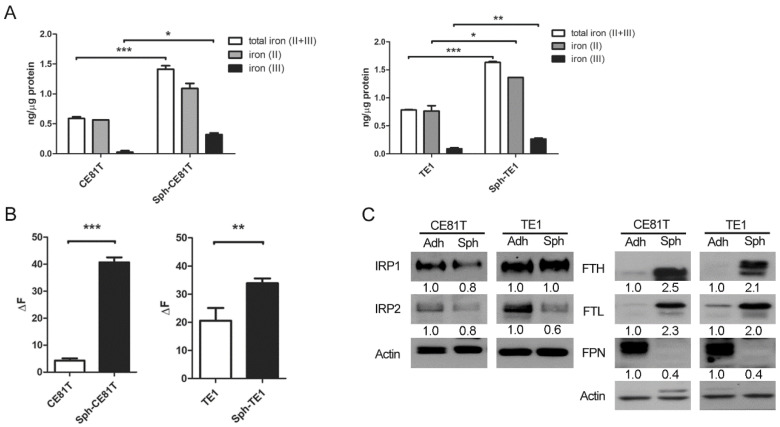
Esophageal cancer cells in spheroid culture have h iron content. (**A**) Total iron assay of esophageal squamous cell carcinoma (ESCC) cells lines, CE81T and TE1, in spheroid (Sph) or adherent (Adh) culture. (**B**) Labile iron pool (LIP) of cells in spheroid and adherent culture. (**C**) Left panel, Western blot analysis with quantification of iron regulatory proteins (IRP) 1 and 2. Right panel, Western blot analysis with quantification of ferritin heavy chain (FTH), ferritin light chain (FTL), and ferroportin (FPN) of cells in spheroid and adherent culture. * *p* < 0.05, ** *p* < 0.01, *** *p* < 0.001 versus control as determined by Student’s *t*-test.

**Figure 2 biomolecules-12-00048-f002:**
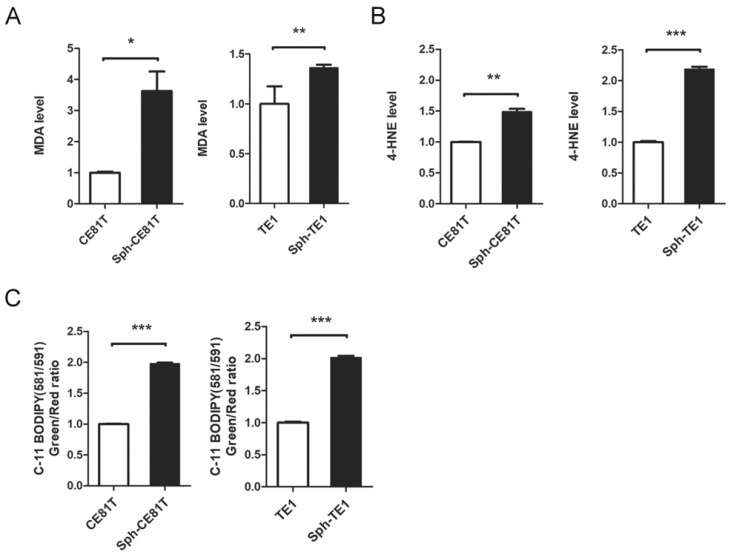
Esophageal cancer cells in spheroid culture have greater lipid peroxidation. (**A**) MDA assay of CE81T and TE1 cells in spheroid or adherent culture. (**B**) 4-HNE assay of cells in different cultures. (**C**) Quantification of C-11 BODIPY staining of cells in different cultures. * *p* < 0.05, ** *p* < 0.01, *** *p* < 0.001 versus control as determined by Student’s *t*-test.

**Figure 3 biomolecules-12-00048-f003:**
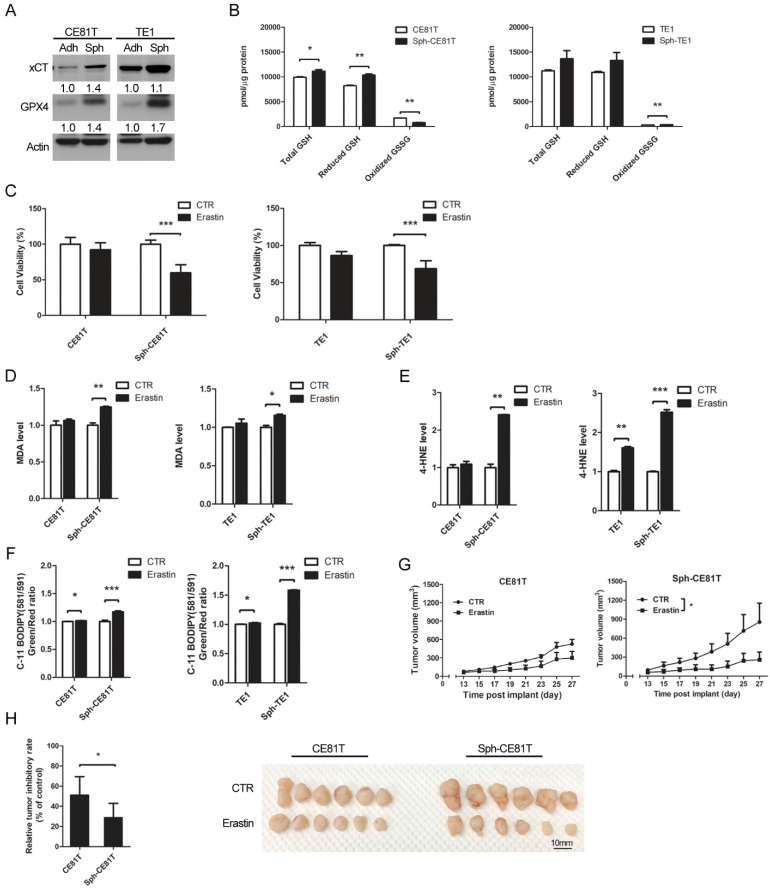
Inhibition of xCT induced more ferroptosis in esophageal cancer cells in spheroid culture. (**A**) Western blot analysis with quantification of xCT and GPX4 in CE81T and TE1 cells in spheroid or adherent culture. (**B**) GSH/GSSH ratio of cells in different cultures. (**C**) Viability of cells in different cultures with or without erastin (20 μM) treatment. (**D**) MDA production of cells in different cultures with or without erastin (20 μM). (**E**) 4-HNE of cells in different cultures with or without erastin (20 μM) treatment. (**F**) Quantification of C-11 BODIPY staining of cells in different cultures with or without erastin (20 μM) treatment. (**G**) Tumor volume and (**H**) weight measurement with photography after subcutaneous injection of 10^6^ CE81T cells in NOD/SCID mice followed by erastin treatment (40 mg/kg, intraperitoneal injection, twice every other day for 2 weeks). * *p* < 0.05, ** *p* < 0.01, *** *p* < 0.001. *p* value of B, G, H were calculated by two-way ANOVA. *p* value of C, D, E, F were calculated by Student’s *t*-test.

**Figure 4 biomolecules-12-00048-f004:**
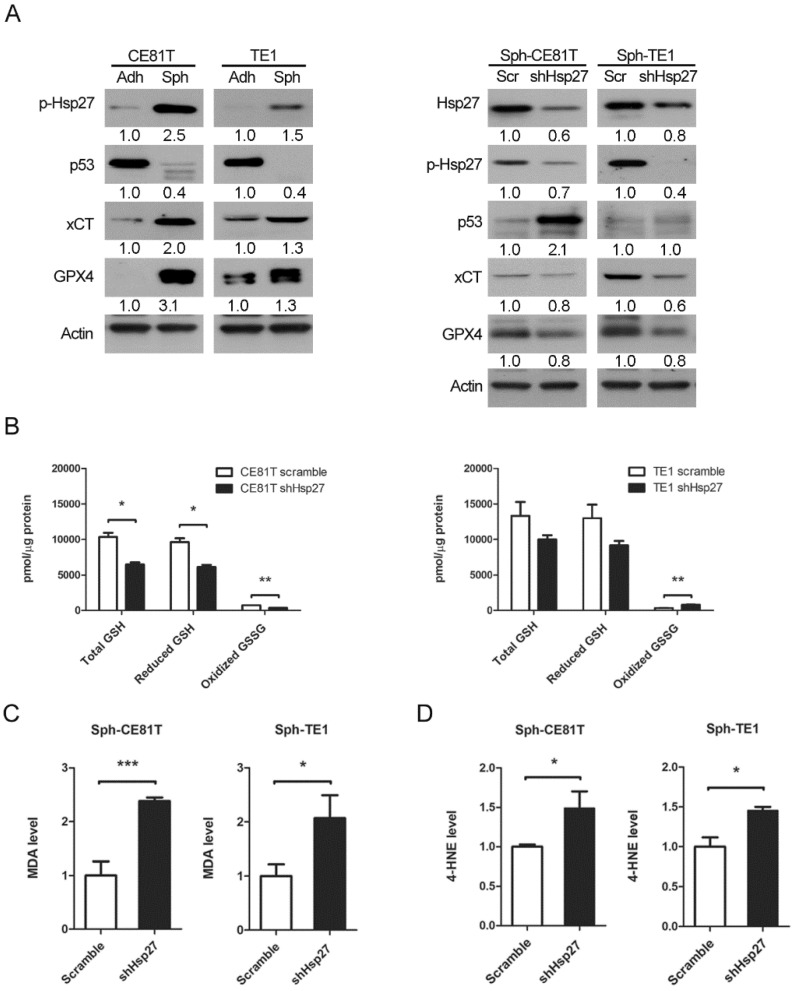
Knockdown of Hsp27 in esophageal cancer cells in spheroid culture induced more lipid peroxidation. (**A**) Left panel, Western blot analysis with quantification of Hsp27, phospho-Hsp27, p53, xCT, and GPX4 of CE81T and TE1 cells in spheroid and adherent culture. Right panel, Western blot analysis with quantification of Hsp27, phospho-Hsp27, p53, xCT, and GPX4 of CE81T and TE1 cells with Hsp27 knockdown (shHsp27) in spheroid culture. Scr, scramble. (**B**) Glutathione measurement of TE1 and CE81T cells with Hsp27 knockdown in spheroid culture. (**C**) MDA production of TE1 and CE81T cells with Hsp27 knockdown in spheroid culture. (**D**) 4-HNE of TE1 and CE81T cells with Hsp27 knockdown in spheroid culture. * *p* < 0.05, ** *p* < 0.01, *** *p* < 0.001. *p* value of B was calculated by two-way ANOVA. *p* value of C and D were calculated by Student’s *t*-test.

**Figure 5 biomolecules-12-00048-f005:**
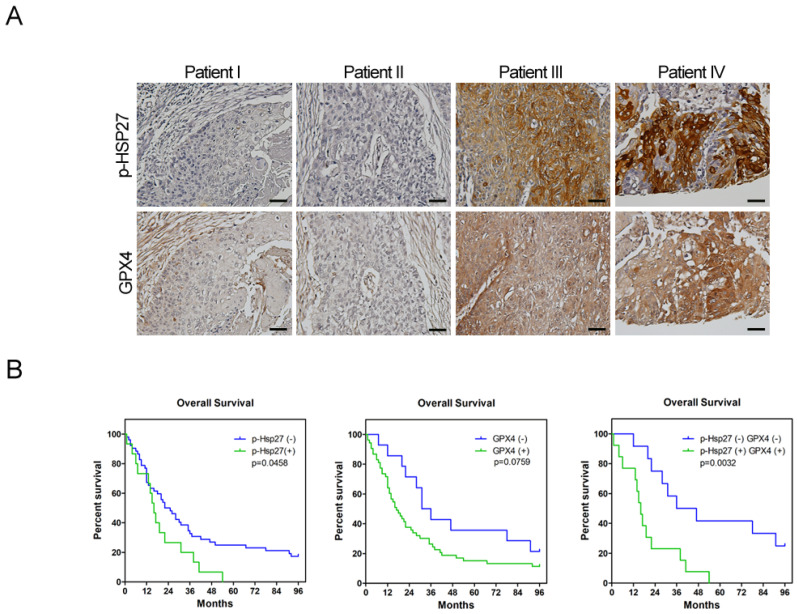
Phospho-Hsp27-GPX4 expression is correlated with prognosis in esophageal cancer patients. (**A**) Immunohistochemical staining of phospho-Hsp27 and GPX4 in patients with esophageal squamous cells carcinoma. (**B**) The survival curves of esophageal cancer patients with or without upregulated phospho-Hsp27 and GPX4 expression (calculated using the Kaplan–Meier method). Expression of phospho-Hsp27 or GPX4 of ≥30% was regarded as positive. Scale bars, 50 μm.

## Data Availability

Not applicable.

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
