# Peer review of "Esophageal Cancer Stem-like Cells Resist Ferroptosis-Induced Cell Death by Active Hsp27-GPX4 Pathway"

_biomolecules, 2021, doi:10.3390/biom12010048_

Round 1

Reviewer 1 Report

The manuscipt of Liu CC and Coll. studied the role of ferroptosis induced-cell death in cancer stem cells (CSC), responsible for tumor initiation and treatment failure in esophageal cancer. They demonstrated  a protective mechanism against ferroptosis through up-regulation of GPX4 and xCT by Hsp27 in CSC and a correlation between phospho-Hsp27 and GPX4 expression levels with poor prognosis in patients with esophageal cancer. They conclude that  targeting Hsp27 or GPX4 to block this intrinsic protective mechanism against ferroptosis is a potential treatment strategy for eradicating CSC in esophageal squamous cell carcinoma.

Albeit the role of ferroptosis and Hsp27 in esophageal cancer has been recently described and  the function of GPX4 in preventing ferroptosis  is already known, however the interaction between Hsp27 and GPX4 represent a novelty.  Overall I find the manuscript interesting, experiments well described and well written. Conclusions are well supported by the experiments.

There are only minor suggestions:

  • To mention references for the two cell lines;
  • As new interesting papers on the role of ferroptosis in esophageal cancer have been just published, I suggest to add at least one of these references (Zhu L,et al  Identification the ferroptosis-related gene signature in patients with esophageal adenocarcinoma. Cancer Cell Int. 2021 Feb 18;21(1):124.; Shi ZZ, et al. Prognostic and Immunological Role of Key Genes of Ferroptosis in Pan-Cancer. Front Cell Dev Biol. 2021 Oct 13;9:748925. ; Lu T,et al. Systematic profiling of ferroptosis gene signatures predicts prognostic factors in esophageal squamous cell carcinoma. Mol Ther Oncolytics. 2021 Feb 20;21:134-143.)
  • To perform densitometric analysis for western blots.
  • To correct legend of black columns of graphics in fig. 4b
  • To increase the dimension of the graphic legends in fig.3
  • To write the whole name of the title of the paragraph 2.6: 4-HNE assay

Reviewer 2 Report

Liu et al. demonstrated the Hsp27-GPX4 axis as a survival pathway in esophageal cancer stem cells against ferroptosis. The study was nicely designed and the authors made a great effort to present how cancer stem cells in the spheroid culture is rescued through the Hsp27 activation and how this phenomenon is indicative to predict the prognosis of esophageal cancer patients. However, a few more experiments might be necessary to validate the spheroid culture system in the capability to enrich cancer stem cells. And perhaps, it’s worth to discuss whether hypoxia in the spheroid system can lead to the observed physiological changes, such increased iron absorption and high ROS.

A few points to consider:

  1. To confirm the enrichment of cancer stem cells from the spheroid culture, you could do a qPCR or immunofluorescence to quantify the expression level of a few stemness markers.
  2. Hypoxia may also affect iron metabolism and cause elevated iron absorption. Are HIF expression levels increased in the spheroid?
  3. Could it be that the ROS in spheroids is also a result of hypoxia?

Author Response

Please see the attachment. Thanks !
